# The Multidimensional Daily Diary of Fatigue-Fibromyalgia-17 Items (MDF-Fibro-17): Evidence from Validity, Reliability and Transcultural Invariance between Portugal and Brazil

**DOI:** 10.3390/jcm9082330

**Published:** 2020-07-22

**Authors:** Marcos C. Álvarez, Maria Luiza L. Albuquerque, Henrique P. Neiva, Luís Cid, Filipe Rodrigues, Diogo S. Teixeira, Diogo Monteiro

**Affiliations:** 1Department of Sport Sciences, University of Beira Interior (UBI), 6201-001 Covilhã, Portugal; marcos.alvarez@ubi.pt (M.C.Á.); luiza.albuquerque@ubi.pt (M.L.L.A.); hpn@ubi.pt (H.P.N.); 2Research Center in Sport, Health and Humam Development (CIDESD), 5001-801 Vila Real, Portugal; luiscid@esdrm.ipsantarem.pt (L.C.); frodrigues@esdrm.ipsantarem.pt (F.R.); 3Sport Science School of Rio Maior, Polytechnique Institute of Santarém (ESDRM-IPSantarém), 2040-413 Rio Maior, Portugal; 4Life Quality Research Center (CIEQV), 2040-413 Rio Maior, Portugal; 5Faculty of Physical Education and Sport (ULHT/FEFD), University of Lusófona, 1749-024 Lisbon, Portugal; diogo.teixeira@ulusofona.pt; 6Center for the Study of Human Performance (CIPER), 1495-751 Lisbon, Portugal

**Keywords:** fibromyalgia, fatigue, questionnaire, cross-cultural validity, measurement invariance

## Abstract

The Multidimensional Daily Diary of Fatigue-Fibromyalgia-17 (MDF-fibro-17) is an instrument that measures the different components of fibromyalgia-related fatigue symptoms. The current study aims to examine the factor structure of the MDF-fibro-17 in a sample of Portuguese and Brazilian patients diagnosed with fibromyalgia. Additionally, a cross-cultural analysis was carried out on these samples to understand the multidimensional complexity of examining the different dimensions of fatigue in patients with different cultural backgrounds and how fibromyalgia impacts patients with this syndrome. A confirmatory factor analysis was performed to examine the psychometric properties of the measure. Additionally, a multigroup analysis was carried out on the samples of these two cultures to examine measurement invariance. In total, 209 Portuguese women aged between 21 and 75 years (M = 47.44; SD = 10.73) and 429 Brazilians women aged between 16 and 77 years (M = 46.51; SD = 9.24) participated in this study. The results revealed that the measurement model provided an acceptable fit to the data in both the Portuguese and Brazilian samples, also displaying acceptable convergent and discriminant validity. In addition, the model showed acceptable internal consistency and was invariant between cultures. In sum, the MDF-fibro-17 is a valid measure that offers a unique assessment of fatigue symptoms in Portuguese and Brazilian women with fibromyalgia.

## 1. Introduction

Fibromyalgia Syndrome (FM) is defined by a chronic and neurological condition that causes sensory changes and is associated to unexplained widespread musculoskeletal pain without the impact of external or internal stimuli (e.g., inflammation) [1,2]. That is, individuals with FM report that they experience some sort of pain in specific parts of the body (e.g., arms, legs and trunk), but the muscles are not affected, and the pain does not originate there, being a sensory response to the FM syndrome. One of the characteristics of FM is the diversity of symptoms presented by individuals affected by this syndrome, but the most reported symptoms are: unexplained muscle pain, excessive fatigue, a decrease in the maximal muscle strength and psychological issues like sleep disorders, anxiety, depression, dizziness, headache and nausea [2,3]. According to the existing literature, FM syndrome has an impact on 2% to 4% of the world population, where middle-aged (i.e., between 30 and 50 years old) women seem to be the group with the greatest incidence [4,5].

Despite the unexplained muscle pain (e.g., inflammation) individuals feel during their daily life, fatigue proves to be one of the main disabling factors for patients with FM. Specifically, patients with FM report that their fatigue is characterized by excessive physical and cognitive tiredness and is not usually eased after hours of sleep or rest, which ends up making it the biggest obstacle for them to overcome physical inactivity and to perform daily tasks [6,7,8]. Thus, the assessment of fatigue helps patients grade the severity of the symptoms of FM, and it is used as a reference when they are asked about their well-being and quality of life [6,9,10].

### 1.1. Fibromyalgia and Fatigue

Symptoms of fatigue can be measured using self-reported measures, in which the patients make a critical and subjective analysis of their perception of this symptom. One of the existing instruments is the Multidimensional Fatigue Inventory (MFI), which has been used in the past to measure subjective fatigue [11]. However, this measure has only been validated in patients with chronic fatigue, college students in psychology and medicine and army recruits, none of which were diagnosed with FM. Additionally, the focus group of the scale were individuals with chronic fatigue not related to any specific medical condition (e.g., FM). Thus, due to the lack of valid instruments for measuring fatigue in patients with FM, Morris and colleagues [10,12] developed the Multidimensional Daily Diary of Fatigue-Fibromyalgia-17 (MDF-fibro-17). The MDF-fibro-17 was created to measure and evaluate the different components of FM-related fatigue, so that it could be possible to understand the general and specific complexity of fatigue in patients with FM [12]. The questionnaire comprises 17 items that measure five dimensions of fatigue in patients with FM, namely: (i) Global fatigue experience; (ii) Cognitive fatigue; (iii) Physical fatigue; (iv) Motivation; and (v) Impact on function. This subdivision is due to the recognition and acceptance of fatigue as a multidimensional factor and to the need for investigation and treatment for each specific dimension of fatigue [10].

Looking deeper at each dimension, global fatigue experience captures certain general points of fatigue which are not suitable for other, more specific domains, as patients with FM use a variety of terms to describe their symptoms of fatigue. This dimension is paramount, since several global terms have been used involving important and comprehensive elements to capture the experience of fatigue in a global manner [10,13]. FM patients also report that their fatigue affects them both physically and mentally [10,14]. With this in mind, there are cognitive limitations described by patients with FM, such as mental tiredness, which end up impacting the concentration to perform specific tasks, to think clearly or to remember something. Thus, the cognitive fatigue dimension examines the extent to which the experience of fatigue limits cognition in patients with FM [10,14]. Patients with FM report that physical fatigue is one of their biggest barriers and problems, mentioning muscle weakness and a feeling of heaviness in their body [10,14]. The physical fatigue dimension measures the extent to which patients with FM experience physical fatigue [10,14]. Patients with FM describe motivation as a direct and integral factor of fatigue, as they have severe difficulty in motivating themselves to perform any physical activity, or even an activity of their daily routine. The motivation dimension is used to check the extent to which motivation (or lack of it) hinders patients with FM to act upon a given activity or behavior [8,10]. At last, patients with FM also describe that fatigue influences their functional capacity in their daily activities. The impact on the function domain evaluates, in a specific manner, how fatigue impacts the performance of basic functions and activities [8,10].

### 1.2. Past Literature and Current Research

As previously stated, several studies have shown that fatigue is one of the main reported symptoms by patients with FM, when questioned about the determinant factor that impacts their general health and overall perception of the FM syndrome [9,13]. Even though fatigue has a major impact on well-being, fatigue is not always included and assessed in clinical research on patients with FM [10,15]. One of the possible reasons for this is the lack of valid measures examining fatigue in patients with FM. As a consequence, the effects of treatments in patients with FM end up not being accurate or conclusive, as has been shown in the literature [16]. For this reason, the MDF-fibro-17 was developed to capture aspects of global fatigue as well as specific dimensions of fatigue, as previously described [10]. To the best of our knowledge, the MDF-fibro-17 has only been validated in American, German and French patients with FM. Additionally, the scale was tested in each sample independently, lacking the validity of measurement invariance, as there could be cultural differences. Thus, the MDF-fibro-17 lacks cultural variance, since multigroup analysis allows scholars to assess the equivalence of the measurement model between groups with different characteristics [17] and to demonstrate whether there are differences between the perception of fatigue symptoms in individuals with FM from different cultural backgrounds. Hence, the process of scale validation is an essential factor when applying the measure in populations with different characteristics [18,19,20,21].

In order to examine the psychometric proprieties of a measure, some criteria must be respected, namely: the sample size should be adequate to the model specification and the item meanings should not suffer differences between groups [17]. In addition, it is important, in scale validation studies, to verify the possible differences in item meanings in groups with different characteristics [18,19,20,21].

In Portugal, the prevalence of FM is between 1.3% to 2.1% of the total population, that is, more than 200,000 adults have been diagnosed with FM [22]. Women are more affected by FM compared to men, with a proportion of six cases compared to one case, respectively [23]. Additionally, the study by Branco et al. [24] showed that Portugal has more patients with FM (mainly women) and that Portuguese patients display greater scores in FM-related symptoms when compared to other European countries (e.g., Spain, France, Germany and Italy). In Brazil, it is estimated that 2% of the population has been diagnosed with FM. Specifically, almost 4.2 million Brazilians suffer from this syndrome, and women are also more affected by FM compared to men, with a proportion of 5.5 cases compared to one case, respectively [25]. These findings support the previously mentioned literature showing that FM predominately impacts women, compared to men, and that Portuguese native speakers have higher scores on FM-related symptoms such as pain and fatigue.

Considering the dynamic and continuous process of instrument examination, and the existing limitations in the literature, the present study aimed to translate and validate the MDF-fibro-27 in a sample of Portuguese and Brazilian patients with FM. Several authors have described the importance of performing measurement invariance analysis to determine whether the scale can be applied to groups with different characteristics [26,27]. Hence, to further examine the validity of the translated measure, a multigroup analysis was performed on the measure to examine a possible invariance according to the cultures under analysis.

## 2. Experimental Section

### 2.1. Participants

Two independent samples were collected for the present study. Sample 1 consisted of 290 Portuguese women aged between 21 and 75 years (M = 47.44; SD = 10.73), who were invited to participate in the study. The Portuguese participants had been diagnosed with FM 7.71 ± 6.04 years ago, on average. Sample 2 consisted of 429 Brazilians women aged between 18 and 77 years (M = 46.51; SD = 9.24). The Brazilian individuals had been diagnosed with FM 7.75 ± 6.43 years ago, on average.

The inclusion criteria for this study were: (a) being older than 18 years, (b) having been diagnosed with FM by a medical doctor specialized in FM, and (c) being a woman. Individuals who self-diagnosed with FM, were underaged and/or were male did not met the inclusion criteria for participation in this study.

### 2.2. Procedure: Data Collection

Before data collection, ethical approval was obtained from the ethic and scientific board of the Research Center in Sport, Health and Human Development (CIDESD), under the reference number UID04045/2020. The current study was conducted according to the Helsinki declaration and its later amendments.

Regarding data collection procedures in the Portuguese sample, the National Association against Fibromyalgia and Chronic Fatigue Syndrome (MYOS) was contacted, and the objectives were explained. After approval, MYOS indicated medical doctors specialized in FM, who informed the authors of existing patients with FM. To facilitate data collection in the Portuguese sample, these medical doctors made available to the authors closed Facebook groups, composed only of patients with FM. After contacting them and explaining the objectives of the study, an informed consent was obtained from each participant individually.

Concerning the Brazilian sample, similar procedures were carried out. The Brazilian Fibromyalgia Association (ABRAFIBRO) was contacted, and the objectives were explained. After the approval of this association, medical doctors specialized in FM were approached to aid the researchers with the data collection. These medical doctors contacted patients with FM using their closed Facebook groups, so that the researchers could contact them afterwards. The objectives of the study were explained and an informed consent was obtained from each potential participant individually prior to completing the survey. All individuals participated voluntarily in this study, and the time taken to complete the questionnaire was approximately 15 min.

### 2.3. Instrument

The MDF-fibro-17 [10,12] was used to measure the different dimensions of FM-related fatigue. This 17-item instrument consists of five subscales, namely: global experience of fatigue (four items; e.g., “*How severe was your fatigue today?*”); physical fatigue (three items; e.g., “*How weak did your muscles felt today?*”); cognitive fatigue (four items; e.g., “*How difficult was it for you to concentrate because you were tired today?”);* motivation (three items; e.g., “*How much of an effort was it for you to do things today?*”); and, impact on function (three items; e.g., “*Did you do things more slowly because you were tired today?*”). Participants responded to each item using a 10-point scale ranging from 0 (“not at all”) to 10 (“extremely”). Higher scores indicated greater fatigue severity. Morris et al. [10] examined the validity of the scale and supported the use of the MDF-fibro-17, as it displayed appropriate model fit (CFI = 0.96 and RMSEA = 0.08).

### 2.4. Procedures: Translation of the Questionnaire

The translation of the MDF-fibro-17 from English to Portuguese was done using the committee approach methodology (see Brislin [28]) as suggested by Banville [29]. The process includes five stages, namely:Preliminary Translation: The first stage was carried out by researchers with the help of three Portuguese native bilingual Portuguese-English teachers. The English version of the questionnaire was translated into Portuguese, which resulted in the first draft;First Evaluation Panel: An analysis of the initial version of the MDF-fibro-17 Portuguese version was performed individually by Portuguese specialists from different areas, such as two medical doctors specialist in FM and four research specialists in psychometric instrument testing. The items received slight syntax and semantic modifications as proposed by the revisions and feedback;Second Evaluation Panel: A revised version of the questionnaire was sent again for evaluation to another panel formed by Brazilian specialists in the same categories as the previous ones (i.e., two medical doctors and four research specialists in psychometric validations). This panel examined all the items in the questionnaire and pointed out some small changes, which were accepted and carried out, so that a new version could be used for preliminary testing;Pilot study: The revised questionnaire was answered by a group of 50 patients with FM (22 from Portugal and 28 from Brazil), to determine if all items were clear and understandable. Data from these participants were not considered for psychometric testing of the MDF-fibro-17 nor for test-retest examination;Final revision: two Portuguese and two Brazilian teachers revised the final translated version of the MDF-fibro-17 to identify possible syntax, spelling and grammar issues. No differences were found in the semantic, spelling and syntax of the Portuguese version either by Portuguese teachers or by the Brazilian teachers. For this reason, the same measure in Portuguese from Portugal was applied to both samples.

### 2.5. Statistical Analysis

Descriptive statistics (means and standard deviation) as well as bivariate correlations were calculated for all dimensions related to the MDF-fibro-17. For test-retest reliability, the recommendations by Banville et al. [29] were followed. Based on the probability theory, a sample size of n = 30 approximates a normal distribution and is therefore considered as acceptable and recommended [30]. For this study, data from 40 Portuguese and 40 Brazilian participants were randomly selected for test-retest evaluation. The time between questionnaire administrations was four weeks, as suggested by Banville et al. [29]. Alfa coefficients were considered for internal consistency, adopting values ≥ 0.70 as acceptable [31].

For model assessment, a Confirmatory Factor Analysis (CFA) employing the maximum likelihood estimator in AMOS 23.0 [32] was performed. Measurement model adequacy was verified by the traditional absolute and incremental indices, namely: Comparative Fit Index (CFI), Tucker–Lewis Index (TLI), Standard Mean Root Square Residual (SRMR) and Root Mean Square Error of Approximation (RMSEA), with a confidence interval of 90%. For model adequacy, the following cutoffs suggested by several authors [33,34] were considered: CFI and TLI ≥ 0.90, SRMR and RMSEA ≤ 0.8. Chi-square and degrees of freedom will be displayed for transparency, but not interpreted, as the chi-square is over-sensitive to sample size and model complexity [31].

Internal consistency was examined through composite reliability coefficients, adopting ≥ 0.70 as cutoff [35]. Average Variance Extracted (AVE) was calculated to examine convergent validity, accepting values > 0.50, as proposed by several authors [30,35]. Discriminant validity was achieved when AVE values were greater than the squared correlation across the constructs of the measurement model [35].

### 2.6. Multigroup Analysis

Several authors [27,31] have shown that measurement invariance is crucial, as it determines whether certain measurements can be applied equally to groups with different characteristics [33]. A multigroup analysis between the Portuguese and Brazilian samples was conducted based on author recommendations [27,31,33]. Specifically, two criteria had to be met to achieve measurement invariance: (1) the measurement model should provide an acceptable fit in each sample; (2) configural, metric, scalar and residual invariance criteria should be respected. In this study, invariance criteria were evaluated considering different cutoffs, specifically: for configural invariance, differences in CFI (ΔCFI) should be less than 0.01 [27]; for metric invariance, differences in CFI (ΔCFI) should be less than 0.01, differences in SRMR (∆SRMR) should be less than 0.030 and differences in RMSEA (∆RMSEA) should be less than 0.015; and, for scalar invariance, ΔCFI should be less than 0.01, ∆SRMR should be less than 0.010 and ∆RMSEA should be less than 0.015, as stated in the previous literature [31]. Differences in chi-square and degrees of freedom will be displayed for transparency but not interpreted, since there are no recommended values for measurement invariance considering these two indicators.

## 3. Results

### 3.1. Preliminary Analysis

A preliminary inspection of the data showed no missing values, and no univariate outliers were detected, since the values of skewness and kurtosis values were comprised within cutoffs, revealing no violation of univariate data distribution. Nevertheless, Mardia’s coefficient for multivariate kurtosis exceeded the recommended value in all samples [33]. Consequently, a Bollen–Stine (B-S) bootstrap of 2000 samples was employed for further analysis [36], as it provides a way of imposing the model on the sample data so that bootstrapping is done under the model specification when data is non-normal, as indicated by the Mardia coefficient. Additionally, adding the bootstrap to the model is appropriate for obtaining adjusted p values for the model fit statistic [36]. Thus, a p value below 0.05 should be considered as indicative of an acceptable model fit, taking into account the previously mentioned traditional absolute and incremental indices.

### 3.2. Test-Retest Analysis

As previously mentioned, test-retest analysis was conducted using a sub-sample of the Portuguese and Brazilian sample, considering 40 randomly selected participants. The results showed that the correlations from the responses given to each item in the first and second administrations of the instrument ranged from 0.72 (Item 14) to 0.89 (Item 15) in the Portuguese sub-sample. In the Brazilian sub-sample, item correlations between administrations ranged from 0.71 (Item 16) to 0.87 (Item 12). In this regard, acceptable test-retest correlations were found indicating that the items had a high degree of temporal reliability. Additionally, alpha coefficients provided acceptable internal consistency (ranging from α = 0.70 to α = 0.89), informing that the items measured the proposed dimension. For detailed information, see Table 1 for test-retest results in the Portuguese sub-sample and Table 2 for the Brazilian sub-sample.

### 3.3. Descriptive Statistics, Internal Consistency and Convergent and Discriminant Validity

Table 3 shows the descriptive statistics, internal consistency estimates, AVE scores and squared correlations for all factors under analysis in both the Portuguese and Brazilian samples. The results showed that individuals from both countries presented high mean (i.e., above midpoint) values in all factors. Moreover, there is evidence that all factors have adequate composite reliability values, since composite reliability coefficients were above cutoff (ranging from 0.88 to 0.95). Additionally, the convergent validity criteria were respected in both samples, since AVE scores were above the 0.5 cutoff. Regarding discriminant validity in the Portuguese sample, 8 of the 10 possible comparisons were confirmed. Only the interaction between global fatigue experience and physical fatigue and the interaction between motivation and impact on function did not display discriminant validity in the Portuguese sample.

In the Brazilian sample, discriminant validity was confirmed in 7 of the 10 possible interactions. Discriminant validity was not achieved in the following interactions in the Brazilian sample: between global fatigue experience and physical fatigue; between global fatigue experience and motivation; between physical fatigue and motivation; and between cognitive fatigue and motivation.

### 3.4. Confirmatory Factor Analysis

Results from the measurement model analysis in each group are displayed in Table 4. The current study showed that the CFA model specification provided an acceptable fit in both the Portuguese and the Brazilian samples. In addition, the items presented factor loadings equal to, or greater than, 0.50, explaining at least 25% of the variance of the latent factor. For detailed information on the factor structure of the model, see Figure 1 (Portuguese sample) and Figure 2 (Brazilian sample).

### 3.5. Measurement Invariance

To test the measurement invariance between cultures, the configural model was compared with the metric model, the scalar model and the residual model. Multigroup analysis (see Table 5) revealed that the **Δ**CFI, **Δ**RMSEA and **Δ**SRMR criteria between models were respected. These analyses suggest that invariance remained stable with each subsequent parameter restraint, showing that the model does not differ across cultures due to the sufficient model fit in each model (i.e., metric, scalar and residual).

## 4. Discussion

The aim of the present study was to address an existing gap in the literature, translating the MDF-fibro-17 into Portuguese and validating the measure in a sample of Portuguese and Brazilian patients with FM. As the structure of the original questionnaire was maintained, this new version differs only in the linguistics, representing a reliable and valid measure of fatigue in Portuguese and Brazilian patients with FM. Additionally, the cross-cultural invariance between Portugal’s and Brazil’s samples showed that the version translated into Portuguese was valid and item meanings were retained in both cultures. The development and testing of measures have become an important focus of research among scholars and medical doctors. Overall, the findings support the utility of the MDF-fibro-17 as a method to obtain a reliable assessment of fatigue-related symptoms in individuals with FM from two distinct cultures.

### 4.1. Factorial Validity of the MDF-Fibro-17

The present results suggest that the proposed five-factor solution assessing the dimensions of fatigue according to the original instrument provided an acceptable fit to the data. Specifically, the current findings support the psychometric proprieties of the MDF-fibro-17 in both the Portuguese and Brazilian samples. Item correlations in the test-retest examination ranged from 0.71 to 0.89, as seen in Table 1 and in Table 2. Thus, the findings provide acceptable test-retest correlations (>0.70), indicating that the MDF-Fibro-17 had a high degree of temporal reliability in both versions. Regarding internal consistency, the results of the present research showed that composite reliability coefficients showed acceptable internal consistency [35]. Similar results have been reported elsewhere [12], showing a good degree of reliability of the current translated versions.

All factorial loadings in the Portuguese adapted 17-item version exhibited acceptable factor loadings (>0.50) and loaded their respective factors significantly (*p* < 0.01), following previous assumptions [26,30]. Additionally, the AVE scores in the present study were above the cutoffs, achieving convergent validity in all factors in both samples. These results provide further support for the validity of the MDF-fibro-17 in both the Portuguese and Brazilian samples, as several criteria for acceptable factor structure were respected [30].

When analyzing discriminant validity, some dimensions did not meet the criteria. Specifically, the factors between the global fatigue experience and physical fatigue and between motivation and impact on function displayed some issues in the Portuguese sample. Discriminant validity is a characteristic of a measure that evaluates the ability to discriminate between factors. Thus, it is assumed that some dimensions in the proposed measure could be overlapping. However, the covariance among global fatigue and physical fatigue and the motivation and impact on function were positive and significant. Additionally, items loaded their pre-defined factor significantly, and no cross-loadings were detected. The removal of one of the factors not discriminating did not increase the model fit, and the overall measurement model exhibited an acceptable fit. Thus, both dimensions were retained, to be as parsimonious as the original model. Considering these aspects, the current results suggest that the dimensions that did not discriminate are indeed distinguishable and should be retained [31].

Looking at the Brazilian sample, our results show discriminant validity in 7 of the 10 possible interactions. As in the Portuguese sample, discriminant validity was not respected in the interactions between global fatigue experience and physical fatigue, between physical fatigue and motivation and between cognitive fatigue and motivation. Like the results of the Portuguese sample, the covariance among these dimensions was positive and significant, and the items loaded their pre-defined factor significantly. Since no cross-loadings were detected and dimension removal did not increase the model fit, the dimensions were retained, and the researchers went on to examine other statistical tests for scale validity. Li and colleagues [12] only found discriminant validity issues between the motivation and physical fatigue dimensions. This can be related to the overall perception of fatigue in patients with FM with different cultural backgrounds. In other words, Portuguese and Brazilian patients with FM could perceive and experience fatigue more as a global perception compared to English patients with FM who could look at fatigue as a demotivating factor for engaging in any given behavior or activity. However, this is only speculative, and more studies with other samples from different cultures are warranted to explore the discriminant validity of the factors. Additionally, future studies with exploratory models and bifactor specifications are needed to examine the multidimensionality of the MDF-fibro-17 not only in Portuguese-speaking individuals but also in English-speaking patients with FM. All in all, more intercultural studies, like this one, are paramount to explore in more detail the perception of fatigue and the characteristics inherent in each culture [37,38].

### 4.2. Measurement Invariance

Regarding the measurement invariance of the 17-item model, the present findings demonstrate the equivalence of the instrument between the two samples under analysis. Specifically, the adapted Portuguese and Brazilian versions of the MDF-fibro-17 are conceptualized and understood in the same manner in both cultures. Considering the premises of model invariance analysis, defined in the method section [26,27], results show that: (a) the model of the MDF-fibro-17 was the same for both countries (configural invariance); (b) the factorial weights of the items were equivalent for both countries (invariance of the measure), that is, each item had the same importance regardless of the group; and (c) the results can be compared between the two countries using the same questionnaire (scale invariance) [12]. According to several authors [27,33], residual invariance is optional, as it is very difficult to achieve. Thus, there is linguistic equivalence and operational applicability of this instrument between the two culturally different countries.

## 5. Conclusions

### 5.1. Limitations

The present findings showed that the MDF-Fibro-17 adapted to Portuguese provided an acceptable fit to both samples and has a great capacity to verify the five components of fatigue in patients with FM. However, the current findings should be considered in light of some limitations. First, this research did not directly measure whether the participants were currently involved in therapies or physical activities to control or attenuate the symptoms of fatigue related to the FM syndrome. This factor should be explored in future research, with control groups to verify the effectiveness of certain therapies to improve certain aspects of fatigue in patients with FM and to determine whether their perception could differ according to the intervention type. Second, the current findings cannot be generalized to other countries or contexts, as further research is needed to establish the universality of the scale. Specifically, future studies should examine the factor structure of the MDF-fibro-17 in other groups with different characteristics.

It is worth mentioning that this study was conducted considering only female patients with FM. This decision was made due to the larger impact of FM in women than in men and because the original study [12] did not present sample characteristics, which could bias interpretations in groups with different socio-demographic characteristics. Thus, considering the existing literature, the researchers opted for a conservative approach, assessing fatigue-related symptoms only in female patients with FM. Therefore, this scale should only be applied and considered as valid in male patients with FM after a careful analysis of the factor structure.

Since this was the first study to translate and validate the MDF-fibro-17 in a culture different from the original one, the focus was based on the factor structure. Although we found solid results concerning the reliability and validity of this measure, there are links between fatigue and other FM-related symptoms that should be further investigated. Future research should examine the relationship between fatigue in patients with FM and other fatigue measures, or measures of symptom load, pain or functional impairment. The results could give crucial insights into how FM-related symptoms such as fatigue relate to other symptoms experienced by patients with FM. Considering the measure itself, the MDF-fibro-17 should be used as a daily measure of fatigue in patients with FM. For this reason, The assessment of fatigue in patients with FM should be conducted on a daily basis. Although our test-retest examination displayed temporal stability, this could be attributed to the cross-sectional non-therapeutic design of this study. Additionally, the four weeks criteria for test-test should be revised in forthcoming studies, as differences could emerge with other time point criteria. Nevertheless, the current evidence should serve as guidance in future research, mainly in experimental studies with therapeutic interventions.

### 5.2. Practical Implications

The validated Portuguese version of the MDF-fibro-17 makes a significant contribution to the literature concerning the measurement of fatigue in patients with FM. The instrument complements the limited literature examining the factor structure of the MDF-fibro-17 [12]. In general, the results obtained by this study provide support for the validity of the original MDF-Fibro-17, adding new evidence on the distinctiveness of fatigue-related dimensions in patients with FM. The present study reinforces the importance of assessing fatigue in patients with FM, since it is one of the main barriers for physical and mental activities.

In conclusion, the results reported provide support for the MDF-fibro-17 in a sample of Portuguese and Brazilian patients with FM. The instrument showed to be multidimensional, detecting five dimensions of fatigue symptoms. The present work reinforced the importance of examining fatigue in patients with FM, demonstrating that this measure is reliable and valid for measuring distinct aspects of fatigue in Portuguese and Brazilian women with FM.

## Figures and Tables

**Figure 1 jcm-09-02330-f001:**
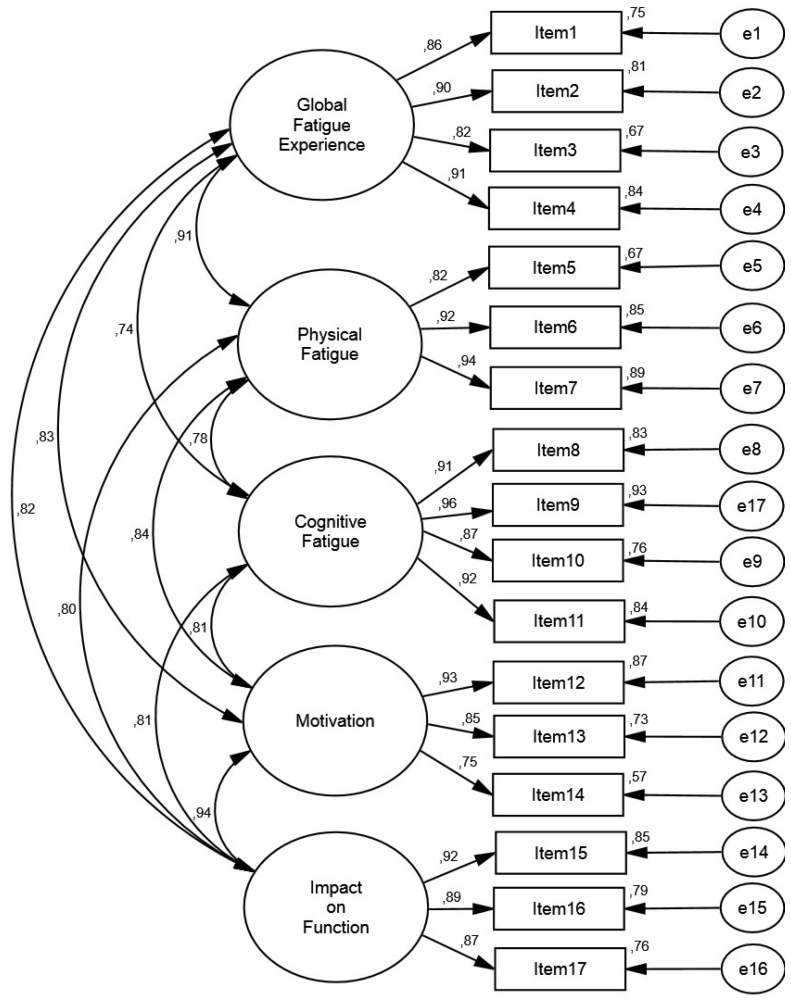
Measurement model in the Portuguese sample.

**Figure 2 jcm-09-02330-f002:**
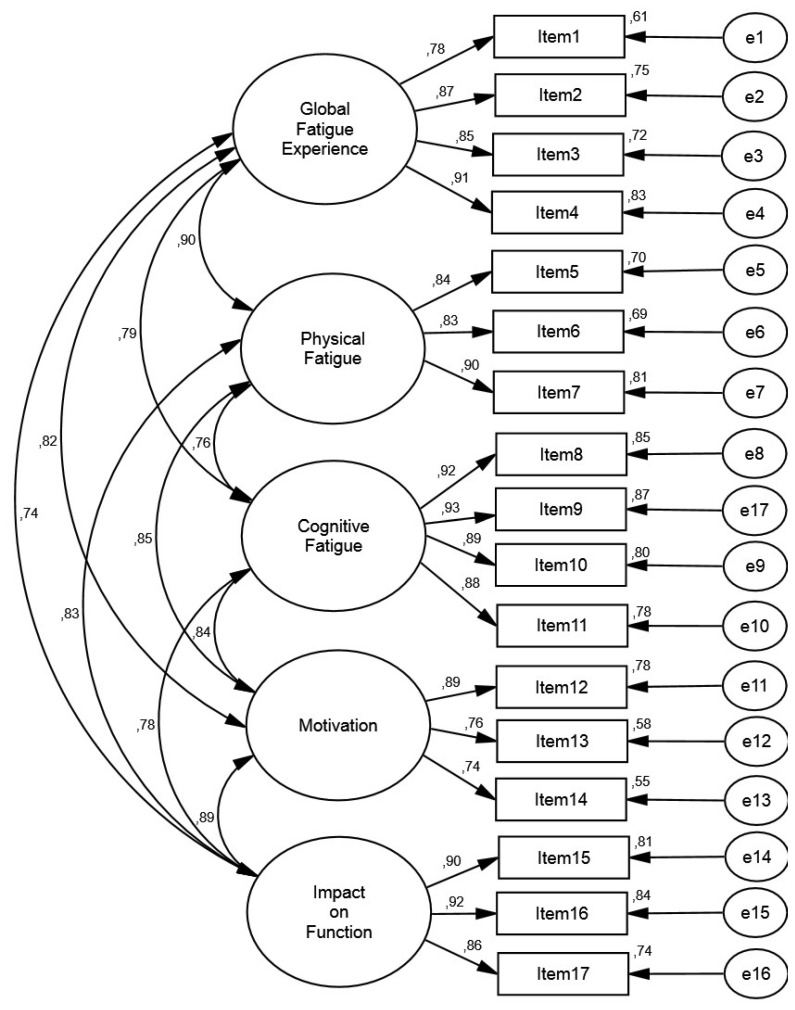
Measurement model in the Brazilian sample.

**Table 1 jcm-09-02330-t001:** Test-retest reliability analysis (Portuguese sub-sample = 40).

Items	M ± SD	*r*	*p*	Alpha
Item 1 Pre-Post	7.24 ± 1.62–7.38 ± 1.77	0.83	<0.001	
Item 2 Pre-Post	7.24 ± 1.68–7.72 ± 1.91	0.82	<0.001	
Item 3 Pre-Post	7.88 ± 1.34–7.82 ± 1.56	0.86	<0.001	
Item 4 Pre-Post	7.39 ± 1.71–7.74 ± 1.82	0.76	<0.001	
Item 5 Pre-Post	7.48 ± 1.64–7.46 ± 2.05	0.81	<0.001	
Item 6 Pre-Post	7.82 ± 1.79–8.03 ± 1.86	0.80	<0.001	
Item 7 Pre-Post	7.70 ± 1.55–8.00 ± 1.72	0.81	<0.001	
Item 8 Pre-Post	7.06 ± 2.16–7.33 ± 2.02	0.78	<0.001	
Item 9 Pre-Post	7.06 ± 2.20–6.23 ± 2.04	0.76	<0.001	
Item 10 Pre-Post	7.48 ± 2.03–7.69 ± 1.72	0.71	<0.001	
Item 11 Pre-Post	7.27 ± 1.86–7.36 ± 1.97	0.81	<0.001	
Item 12 Pre-Post	7.79 ± 1.45–7.90 ± 1.68	0.84	<0.001	
Item 13 Pre-Post	7.89 ± 2.05–7.67 ± 2.19	0.76	<0.001	
Item 14 Pre-Post	7.45 ± 2.18–7.56 ± 2.43	0.72	<0.001	
Item 15 Pre-Post	7.64 ± 1.64–7.77 ± 1.95	0.89	<0.001	
Item 16 Pre-Post	7.64 ± 1.95–7.51 ± 2.45	0.74	<0.001	
Item 17 Pre-Post	7.85 ± 1.73–7.82 ± 2.32	0.86	<0.001	
Global Fatigue Experience Pre-Post	7.44 ± 1.47–7.67 ± 1.58	0.87	<0.001	0.77–0.78
Physical Fatigue Pre-Post	7.67 ± 1.47–7.83 ± 1.75	0.76	<0.001	0.80–0.77
Cognitive Fatigue Pre-Post	7.22 ± 1.94–7.40 ± 1.83	0.84	<0.001	0.81–0.83
Motivation Pre-Post	7.70 ± 1.61–7.71 ± 1.74	0.81	<0.001	0.82–0.81
Impact on Function Pre-Post	7.71 ± 1.67–7.71 ± 2.18	0.74	<0.001	0.78–0.77

M = Mean; SD = Standard Deviation; r = bivariate correlations; *p* = level of significance.

**Table 2 jcm-09-02330-t002:** Test-retest reliability analysis (Brazilian sub-sample = 40).

Items	M ± SD	*r*	*p*	Alpha
Item 1 Pre-Post	6.39 ± 2.43–7.03 ± 2.34	0.80	<0.001	
Item 2 Pre-Post	6.70 ± 2.16–7.32 ± 2.17	0.86	<0.001	
Item 3 Pre-Post	7.04 ± 2.08–7.46 ± 2.18	0.80	<0.001	
Item 4 Pre-Post	7.00 ± 2.52–7.59 ± 2.41	0.77	<0.001	
Item 5 Pre-Post	7.43 ± 2.31–7.78 ± 2.12	0.84	<0.001	
Item 6 Pre-Post	7.78 ± 2.28–8.05 ± 2.05	0.83	<0.001	
Item 7 Pre-Post	7.74 ± 2.09–8.11 ± 1.98	0.84	<0.001	
Item 8 Pre-Post	6.70 ± 2.34–7.51 ± 2.24	0.80	<0.001	
Item 9 Pre-Post	6.65 ± 2.34–7.38 ± 2.67	0.79	<0.001	
Item 10 Pre-Post	6.17 ± 2.64–7.14 ± 2.58	0.72	<0.001	
Item 11 Pre-Post	6.43 ± 2.57–7.19 ± 2.45	0.83	<0.001	
Item 12 Pre-Post	7.39 ± 2.25–7.89 ± 2.12	0.87	<0.001	
Item 13 Pre-Post	7.43v1.95–7.89 ± 2.05	0.75	<0.001	
Item 14 Pre-Post	7.22 ± 2.76–7.54 ± 2.56	0.73	<0.001	
Item 15 Pre-Post	7.35 ± 1.99–7.95 ± 1.94	0.88	<0.001	
Item 16 Pre-Post	7.00 ± 2.45–7.76 ± 2.35	0.71	<0.001	
Item 17 Pre-Post	7.26 ±2.40–7.92 ± 2.29	0.84	<0.001	
Global Fatigue Experience Pre-Post	6.78 ±2.06–7.35 ± 2.05	0.88	<0.001	0.76–0.74
Physical Fatigue Pre-Post	7.65 ± 2.10–7.98 ± 1.44	0.79	<0.001	0.81–0.79
Cognitive Fatigue Pre-Post	6.49 ± 2.37–7.30 ± 2.30	0.86	<0.001	0.79–0.78
Motivation Pre-Post	7.35 ± 2.10–7.77 ± 2.03	0.85	<0.001	0.80–0.81
Impact on Function Pre-Post	7.20 ± 2.18–7.87 ± 2.12	0.77	<0.001	0.78–0.76

M = Mean; SD = Standard Deviation; r = bivariate correlations; *p* = level of significance.

**Table 3 jcm-09-02330-t003:** Descriptive statistics, composite reliability coefficients, AVE scores and squared correlations.

Samples	M	SD	CR	AVE	r^2^				
*Portuguese*					1	2	3	4	5
(1) Global Fatigue Experience	7.25	1.58	0.93	0.77	1	-	-	-	-
(2) Physical Fatigue	7.49	1.64	0.92	0.80	0.83	1	-	-	-
(3) Cognitive Fatigue	7.20	1.85	0.95	0.84	0.55	0.59	1	-	-
(4) Motivation	7.50	1.70	0.88	0.72	0.69	0.71	0.65	1	-
(5) Impact on Function	7.45	1.83	0.92	0.80	0.67	0.65	0.65	0.89	1
*Brazilian*									
(1) Global Fatigue Experience	7.72	1.73	0.92	0.73	1	-	-	-	-
(2) Physical Fatigue	8.22	1.58	0.89	0.73	0.80	1	-	-	-
(3) Cognitive Fatigue	7.88	1.82	0.95	0.83	0.63	0.58	1	-	-
(4) Motivation	8.19	1.71	0.84	0.64	0.67	0.72	0.70	1	-
(5) Impact on Function	8.26	1.80	0.92	0.80	0.55	0.70	0.61	0.80	1

M = Mean; SD = Standard Deviation; CR = Composite Reliability; AVE = Average Variance Extracted; r^2^ = squared correlation.

**Table 4 jcm-09-02330-t004:** Goodness-of-fit indexes of the Portuguese and Brazilian versions and the original model of the Multidimensional Daily Diary of Fatigue-Fibromyalgia-17 (MDF-fibro-17).

Models	χ2	df	B-S p	CFI	TLI	SRMR	RMSEA	RMSEA CI90%
MDF-fibro-17- PT version	369.21	109	<0.001	0.954	0.943	0.030	0.080	0.076–0.082
MDF-fibro-17- BR version	381.48	109	0.002	0.962	0.953	0.026	0.076	0.068–0.085
MDF-fibro-17- Original version	213.43	109	-	0.950	0.930	0.025	0.071	0.109–0.186

χ² = chi-squared; df = degrees of freedom; B-S p = Bollen–Stine bootstrap; CFI = Comparative Fit Index; TLI = Tucker–Lewis Index; SRMR = Standardized Root Mean Square Residual; RMSEA = Root Mean Squared Error of Approximation; CI = Confidence Interval at 90%; PT = Portugal; BR = Brazil.

**Table 5 jcm-09-02330-t005:** Goodness-of-Fit Indexes of the multigroup analysis between the Portuguese and Brazilian samples.

Model	χ2	df	Δχ2	Δdf	p	CFI	ΔCFI	RMSEA	ΔRMSEA	SRMR	ΔSRMR
Configural Invariance	750.94	218	-	-	-	0.959	-	0.058	-	0.030	
Metric Invariance	769.69	230	18,755	12	0.095	0.958	0.001	0.057	0.001	0.029	0.009
Scale Invariance	817.25	245	66,313	27	<0.001	0.956	0.003	0.057	0.001	0.034	0.004
Residual Invariance	968.21	262	217.27	44	<0.001	0.945	0.014	0.061	0.003	0.045	0.015

χ² = chi-squared; df = degrees of freedom; ∆χ² = differences in chi-squared; ∆df = differences in df; CFI = Comparative Fit Index; ∆CFI = differences in CFI; RMSEA = Root Mean Square Error of Approximation; ∆RMSEA = differences in RMSEA; SRMR = Standardized Root Mean Square Residual; ∆SRMR = difference in SRMR.

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
