# Peer review of "The Multidimensional Daily Diary of Fatigue-Fibromyalgia-17 Items (MDF-Fibro-17): Evidence from Validity, Reliability and Transcultural Invariance between Portugal and Brazil"

_jcm, 2020, doi:10.3390/jcm9082330_

Round 1

Reviewer 1 Report

General comments:

Generally well conducted paper, concise and informative. Issues with language at points, needs to be proof-read by an English native speaker. Minor errors throughout (e.g., incorrect tense forms as on p4l140 "exceeded" or p5l163 "display". Methodologically careful and diligent work. A very commendable aspect is the use of a cross-cultural cohort of Portugese speakers. Also, the appendix isn't being referred to in-text, somewhat unclear what the idea is here?

Specific comments:

p1/2 - the MDF should be mentioned in the introduction as it is the primary focus of the paper. Also, the last sentences of the intro are too long and confusingly written.

p2l63 - "Sample 2"

p2l66 - not clear how facebook was used and what doctors were doing of facebook?

Author Response

Reviewer: 1

General comments:

Generally, well conducted paper, concise and informative.

R: We appreciate your time and effort in the review process.

Issues with language at points, needs to be proof-read by an English native speaker. Minor errors throughout (e.g., incorrect tense forms as on p4l140 "exceeded" or p5l163 "display".

R: Substantial revisions were made in the English language writing style. Methodologically careful and diligent work. A very commendable aspect is the use of a cross-cultural cohort of Portuguese speakers. Also, the appendix isn't being referred to in-text, somewhat unclear what the idea is here?

R: The appendix was removed.

Specific comments:

p1/2 - the MDF should be mentioned in the introduction as it is the primary focus of the paper. Also, the last sentences of the intro are too long and confusingly written.

R: The introduction section was substantially revised.

p2l63 - "Sample 2"

R: We appreciate your observation. The number 2 was inserted.

p2l66 - not clear how facebook was used and what doctors were doing of facebook?

R: We apologize for not making clear the data collection procedures. Detailed information was added on how data were collected in the Portuguese and Brazilian sample

Reviewer 2 Report

The Multidimensional Daily Diary of Fatigue-Fibromyalgia 17 Items (MDF-Fibro-17): Evidence from Validity, Reliability and Transcultural Invariance between Portugal and Brazil

This study introduces the Portuguese version oft he MDF-Fibro-17, investigates its reliability (re-test and internal consistency) and factor structure as well as discriminant validity of the factors in a sample of Portuguese women (n=209, 40 for re-test) and in a sample of Brazilian women (n=429, 40 for re-test). Moreover, the authors investigated transcultural invariance between the Portuguese and the Brazilian sample. Overall, the study shows satisfactory results in the aforementioned indices.

Broad comments:

The evaluation and validation of questionnaires in different languages is important before employing the measures in studies. One strength of this study is the investigation of Portuguese as well as Brazilian samples. I also got the impression that the statistics are solid.

However, the manuscript would greatly benefit from proof-reading by an English native speaker. Also, the theoretical background should be revised in order to improve succinctness and stringency. Furthermore, some conclusions should be made more cautious. Specifics see below.

Specific comments:

Abstract:

  • Instead of „explores and evaluates“ I would recommend stating that the questionnaire „measures“
  • I think „is able to measure which domain of fatigue has the greatest impact; thus, clarifying the possible treatments to this disease“ is not a conclusion that can be made from the current study. Please re-think.

Introduction:

  • It seems like, in order to decrease the word count of your introduction, you submitted the main part of your theoretical background as appendix A. Unfortunately, I think that this decreases the readability of the article a lot. Important information like the definition of fatigue and description of the different dimensions of fatigue as assessed by the questionnaire are missing beforehand. Why not actually try to shorten the introduction without outsourcing crucial information into the Appendix?
    • In addition, the Appendix is not mentioned once during the main text of the manuscript. If you stick with outsourcing some information to the Appendix, please reference it accordingly.
  • Other things I noticed in the Introduction part:
  • FM does not „cause“ musculoskeletal pain without external stimuli, but is defined by that. Although significantly overlapping, CFS is a different disorder.
    • Better: FM is defined by… and often accompanied by…
  • What kind of „neurochemical imbalances“ do you mean? Please elaborate.
  • Please indicate which symptoms define FM (and which criteria you used to diagnose FM), because the defining symptoms do not match the most commonly reported symptoms you list
  • Please indicate which questionnaires you mean in reference to Wolfe et al. (2010) – I assume WPI and SSS (measures fatigue, exhaustion at awakening, cognitive symptoms in the last week + depressive symptoms, headaches and gastrointestinal pain in the last 6 months), ACR criteria

2 experimental section

2.1 participants

  • please indicate who diagnosed the participants with which criteria and why you only included women in the study.
  • Please state your exclusion and inclusion criteria.
  • Portuguese and Brazilian samples have the exact same M and SD of illness duration – is that correct or a typo?

2.2 data collection

  • Please introduce the „panel of specialists“ before refering to it in the data collection session
  • Recruitment and data collection should be explained in more detail: How did you choose which Facebook-groups you contacted? Were there any dropouts? Did the doctors hand out the questionnaires and send it back to you or did each participant send her questionnaire back to you individually? etc.

2.3 instrument

  • Again, you should refer the reader to the appendix for a better understanding of what you mean by „different components of FM-related fatigue“ (see also my second comment concerning the Appendix)
  • Did you assess any other questionnaires than the MDF-fibro-17?
  • I think the correct reference in this section is „12“ (Morris et al.), not 10 (Hudson et al.)
  • You state that „previous studies supported validity and reliability“ – please specify the reliability and validity values in the cited study

2.4 translation

  • For people who do not speak Portuguese (like myself), please indicate whether there are any major differences between the Portuguese Portuguese and the Brazilian Portuguese in its written form
  • Please indicate whether the persons involved in the translation and evaluation steps 1-3 were Portuguese or Brazilian

2.5 Statistical Analyses

  • Please specify for which variables you calculated bivariate correlations (i.e. which were the „variables under analysis“?)
  • How did you retrieve the 40 Portuguese and Brazilian subjects from the total sample? Did you choose randomly?
    • Also, please explain why you did not re-test the total sample
  • Were there any dropouts or did you exclude any data before analysis?

Results

3.1 preliminary analysis

  • I am not very familiar with the Bollen-stine bootstrap method, as might other readers not be as well. Can you explain in one sentence, how the indicator in table 4 („B-S p“) is to be interpreted?

3.2 Test-Retest Analysis

  • Tables 1 and 2: As these analyses are based on a sub-sample, please indicate the sample size in the caption of the table
    • Also, please include a description of „Alpha“ within the note.

3.3 descriptive statistics, internal consistency, and convergent and discriminant validity

  • Table 3: Please include a short description of „composite reliability“ within the note of the table or within the text.
  • Last sentence in the paragraph (line 164-166, comparisons that did not achieve discriminant validity): What sample (Portuguese or Brazilian) is this sentence referring to?

3.4

  • Table 4: There are two Chi² values in the first model. Is that correct or a typo?

4. Discussion

  • Line 204/205: In how far is the instrumtent different than the English version?

4.1 Factorial validity

  • Line 214/215: Please rephrase this sentence to clarify that it refers to the test-retest correlations.
  • Line 224/225: „The multidimensional structure…“ – I do not understand this sentence. Could you re-phrase?

4.2 measurement invariance

  • Line 246ff: How is the difference in discriminant validity between the Portuguese and the Brazilian sample to be interpreted? What does the missing discriminant validity of some of the factors mean for the validity of the questionnaire?

5.1 Limitations

  • Please also discuss the implications of not including men in your study
  • Please discuss the issue of construct validity using external measures, because no other fatigue measures, or measures of symptom load, pain measures, measures of functional impairment etc. have been used in your study – Are there any other studies on the same questionnaire (independent of which language was used) that investigated construct validity using external measures? What did they find?

5.2 practical implications

  • The questionnaire is referring to „today“ and called a „diary“. Is it originally designed to measure fatigue symptoms on a day-to-day basis? If so, it should be sensitive to day-to-day changes in fatigue. Also, you suggest using the questionnaire to measure changes in fatigue by therapy etc. – the questionnaire should therefore be sensitive to change. Please discuss this issue in light of the fact that test-retest-reliability, using 2 measurement points 4 weeks apart, was high.

Appendix A:

  • See my first comment referring to the introduction
  • One idea might be to describe the different dimensions of fatigue in greater detail in the „instrument“ section (2.3), as it is hard to understand what the part of the Appendix, where you describe the different dimensions and how they relate to FM symptoms, refers to.
  • Also, I suggest not introducing the MFI in such detail, because you did not assess the MFI, but a different questionnaire.

Author Response

Reviewer: 2

The Multidimensional Daily Diary of Fatigue-Fibromyalgia 17 Items (MDF-Fibro-17): Evidence from Validity, Reliability and Transcultural Invariance between Portugal and Brazil.

This study introduces the Portuguese version of the MDF-Fibro-17, investigates its reliability (re-test and internal consistency) and factor structure as well as discriminant validity of the factors in a sample of Portuguese women (n=209, 40 for re-test) and in a sample of Brazilian women (n=429, 40 for re-test). Moreover, the authors investigated transcultural invariance between the Portuguese and the Brazilian sample. Overall, the study shows satisfactory results in the aforementioned indices.

R: We appreciate your time and effort in the review process.

Broad comments:

The evaluation and validation of questionnaires in different languages is important before employing the measures in studies. One strength of this study is the investigation of Portuguese as well as Brazilian samples. I also got the impression that the statistics are solid.

R: We thank the reviewers for his/her comments.

However, the manuscript would greatly benefit from proof-reading by an English native speaker.

R: Substantial revisions were made in the English language writing style.

Also, the theoretical background should be revised in order to improve succinctness and stringency.

R: The redundancy was also fixed, so that the study was more succinct.

Furthermore, some conclusions should be made more cautious.

R: All possible hasty conclusions were duly corrected.

Specific comments:

Abstract:

  • Instead of „explores and evaluates“ I would recommend stating that the questionnaire „measures“

R: Sentence was revised.

  • I think „is able to measure which domain of fatigue has the greatest impact; thus, clarifying the possible treatments to this disease“ is not a conclusion that can be made from the current study. Please re-think.

R: We thank the reviewer for his/her comment. Sentence was revised.

Introduction:

  • It seems like, in order to decrease the word count of your introduction, you submitted the main part of your theoretical background as appendix A. Unfortunately, I think that this decreases the readability of the article a lot. Important information like the definition of fatigue and description of the different dimensions of fatigue as assessed by the questionnaire are missing beforehand. Why not actually try to shorten the introduction without outsourcing crucial information into the Appendix?

R: The reviewer is right. We mistakenly removed part from the introduction section and moved it to the appendix. Substantial revisions were made in the introduction section and thus appendix A was eliminated.

  • In addition, the Appendix is not mentioned once during the main text of the manuscript. If you stick with outsourcing some information to the Appendix, please reference it accordingly.

R: The appendix was removed.

  • Other things I noticed in the Introduction part:
  • FM does not „cause“ musculoskeletal pain without external stimuli, but is defined by that. Although significantly overlapping, CFS is a different disorder.

R: The Reviewer is correct, the FM does not actually cause chronic fatigue syndrome, but rather excessive fatigue, which makes the wearer feel tired, even with long periods of rest.

  • Better: FM is defined by… and often accompanied by…

R: The definition and the symptoms of FM were revised.

  • What kind of „neurochemical imbalances“ do you mean? Please elaborate.

R: We apologize for not providing a rationale on the associations of neurochemical imbalances. The neurochemical balance is in relation to neurotransmitters. The sentence was revised.

  • Please indicate which symptoms define FM (and which criteria you used to diagnose FM), because the defining symptoms do not match the most commonly reported symptoms you list

R: The list of symptoms has been revised and is in accordance with the proposed and contemporary literature.

  • Please indicate which questionnaires you mean in reference to Wolfe et al. (2010) – I assume WPI and SSS (measures fatigue, exhaustion at awakening, cognitive symptoms in the last week + depressive symptoms, headaches and gastrointestinal pain in the last 6 months), ACR criteria

R: The reviewer is right. The two questionnaires are the WPI and the SSS, which were properly named in the text.

2 experimental section

2.1 participants

  • please indicate who diagnosed the participants with which criteria and why you only included women in the study.

R: The patients who answered the questionnaires were diagnosed by their medical doctors who monitor their disease. The recruitment process was explained in detail in the manuscript. Regarding the inclusion criteria of only female patients with FM, it was based on the existing limitations in the original validation study (Morris et al., 2017), which does not discriminate  the gender of the participants. In addition, since FM has a larger impact on women compared to men, we conducted the study only with female patients with FM.

  • Please state your exclusion and inclusion criteria.

R: Inclusion and exclusion criteria were inserted.

  • Portuguese and Brazilian samples have the exact same M and SD of illness duration – is that correct or a typo?

R: It was a typo. We revised the M and SD in the Brazilian sample.

2.2 data collection

  • Please introduce the „panel of specialists“ before referring to it in the data collection session

R: The reference of “panel of specialists” was related to the procedures of the scale translation. Thus, the sentence was revised for clarification.

  • Recruitment and data collection should be explained in more detail: How did you choose which Facebook-groups you contacted? Were there any dropouts? Did the doctors hand out the questionnaires and send it back to you or did each participant send her questionnaire back to you individually? etc.

R: Data collection procedures were revised in detail for transparency and comprehension. There were no dropouts since the study design was cross-sectional and participation was voluntary.

  • instrument
  • Again, you should refer the reader to the appendix for a better understanding of what you mean by „different components of FM-related fatigue“ (see also my second comment concerning the Appendix)

R: We reiterate that the appendix was removed from the revised manuscript. Sentences were clarified.

  • Did you assess any other questionnaires than the MDF-fibro-17?

R: This is a good point. No, we did not as the purpose of this study was to examine the validity of this instrument in both Portuguese and Brazilian women.

  • I think the correct reference in this section is „12“ (Morris et al.), not 10 (Hudson et al.)

R: Yes, you are right. The citation was revised.

  • You state that „previous studies supported validity and reliability“ – please specify the reliability and validity values in the cited study

R: Sentence was revised.

2.4 translation

  • For people who do not speak Portuguese (like myself), please indicate whether there are any major differences between the Portuguese Portuguese and the Brazilian Portuguese in its written form

R: After final revision of the measure in step 5, two Portuguese and two Brazilian teachers revised the final translated version of the MDF-fibro-17 to identify possible syntax, spelling, and grammar issues. No differences were found in the semantic, spelling, and syntax of the Portuguese version either by Portuguese teachers or by the Brazilian teachers. For this reason, the same measure in Portuguese Portuguese was applied to both samples.

Please indicate whether the persons involved in the translation and evaluation steps 1-3 were Portuguese or Brazilian

R: The researchers (Portuguese and Brazilian native) were part in the preliminary translation processes with 3 Portuguese native bilingual English-Portuguese teachers. In step 2, we contacted Portuguese specialists, and in step 3, we contacted Brazilian specialists.  The Steps in the translation process were clarified in the manuscript.

2.5 Statistical Analyses

  • Please specify for which variables you calculated bivariate correlations (i.e. which were the „variables under analysis“?)

R: Descriptive statistics (means and standard deviation), as well as bivariate correlations were calculated for all dimensions related to the MDF-fibro-17. We inserted this sentence in the statistical analysis.

  • How did you retrieve the 40 Portuguese and Brazilian subjects from the total sample? Did you choose randomly?

R: Sentence was revised. Participants were randomly selected for test-retest reliability.

  • Also, please explain why you did not re-test the total sample

R: This is a good point. We followed Banville et al. suggestions considering at least 30 individuals for test-rest reliability. For this study, we considered 40 Portuguese and Brazilian participants. Considering the total sample is not necessary for this examination of temporal stability as proposed by Vallerand 1989.

  • Were there any dropouts or did you exclude any data before analysis?

R: No, we did not. Since this was a cross-sectional study, the inclusion criteria were defined a priori, and informed consent was obtained before providing participants with the survey, no participants dropped out or were excluded before data analysis.

Results

3.1 preliminary analysis

  • I am not very familiar with the Bollen-Stine bootstrap method, as might other readers not be as well. Can you explain in one sentence, how the indicator in table 4 („B-S p“) is to be interpreted?

R: Explanation was added.

3.2 Test-Retest Analysis

  • Tables 1 and 2: As these analyses are based on a sub-sample, please indicate the sample size in the caption of the table

R: sample size was added

  • Also, please include a description of „Alpha“ within the note.

R: Description was added in the text.

3.3 descriptive statistics, internal consistency, and convergent and discriminant validity

  • Table 3: Please include a short description of „composite reliability“ within the note of the table or within the text.

R: Description was added in the text.

  • Last sentence in the paragraph (line 164-166, comparisons that did not achieve discriminant validity): What sample (Portuguese or Brazilian) is this sentence referring to?

R: Sentence was clarified.

3.4

  • Table 4: There are two Chi² values in the first model. Is that correct or a typo?

R: The reviewer was correct; the second Chi-square value was a typo. We have corrected the value in table 4.

  1. Discussion
  • Line 204/205: In how far is the instrument different than the English version?

R: Sentence was clarified.

4.1 Factorial validity

  • Line 214/215: Please rephrase this sentence to clarify that it refers to the test-retest correlations.

R: Sentence was clarified.

  • Line 224/225: „The multidimensional structure…“ – I do not understand this sentence. Could you re-phrase?

R: Sentence was removed as it did not add up any crucial content to the factor structure examination.

4.2 measurement invariance

  • Line 246ff: How is the difference in discriminant validity between the Portuguese and the Brazilian sample to be interpreted? What does the missing discriminant validity of some of the factors mean for the validity of the questionnaire?

R: Explanation was added. In fact, the discriminant validity discussion as substantially revised.

5.1 Limitations

  • Please also discuss the implications of not including men in your study

R: We appreciate your suggestion. Implications were added and explained.

  • Please discuss the issue of construct validity using external measures, because no other fatigue measures, or measures of symptom load, pain measures, measures of functional impairment etc. have been used in your study – Are there any other studies on the same questionnaire (independent of which language was used) that investigated construct validity using external measures? What did they find?

R: This is a good point. The authors of the original scale did not examine concurrent validity with manifest or other construct-related measures. Additionally, this was the first study to examine the psychometric proprieties of the measure in a culture different from the original. The limitation section was substantially revised according to current results, and those results shown in previous studies considering this measure.

5.2 practical implications

  • The questionnaire is referring to „today“ and called a „diary“. Is it originally designed to measure fatigue symptoms on a day-to-day basis? If so, it should be sensitive to day-to-day changes in fatigue.

R: The reviewer is correct, the original MDF is to be used as a fatigue tester almost daily, so you can measure the temporal buoyancy of fatigue, by test-retests. This is one of the limitations of this work, but it serves as an aid for possible future research, where it is possible to measure fatigue at different times.

Also, you suggest using the questionnaire to measure changes in fatigue by therapy etc. – the questionnaire should therefore be sensitive to change. Please discuss this issue in light of the fact that test-retest-reliability, using 2 measurement points 4 weeks apart, was high.

R: This is a good point. We added a rationale for future assessment of fatigue in patients with FM.

Appendix A:

  • See my first comment referring to the introduction

R: The appendix was removed and content was added to the introduction section.

  • One idea might be to describe the different dimensions of fatigue in greater detail in the „instrument“ section (2.3), as it is hard to understand what the part of the Appendix, where you describe the different dimensions and how they relate to FM symptoms, refers to.

R: As the explanation of the domains of fatigue was moved to the introduction section, the description and interpretation of each dimension were simplified.

  • Also, I suggest not introducing the MFI in such detail, because you did not assess the MFI, but a different questionnaire.

R: We appreciate your concern. We reframed our focus only on the MDF-fibro-17.

Round 2

Reviewer 2 Report

Thank you very much for your consideration of our comments and suggestions. I think the manuscript has improved a great deal by your revision.

Generally, I still think the manuscript should be thoroughly proof-read for English grammar by an English native speaker. The wording still seems off to me and I detected some typos throughout the manuscript.

Content-wise, please make sure that the reader knows that FM is by definition predominantly defined as a disorder of widespread medically unexplained pain. Although fatigue is very important and also part of FM criteria, it is not the main defining symptom.

Also, assessment of tender points has been omitted from the 2010 ACR criteria. If you used those criteria, you should consider not introducing and referring to tender points - or doing so less prominently.

Furthermore, I suggest using the term "persons with FM" or "patients with FM", not "FM carriers" or "patients of the disease". 

Author Response

July, 2020

Dear Dr. Natasa Doronjga

Section Editor

in Journal of Clinical Medicine

Subject: The Multidimensional daily diary of fatigue-fibromyalgia 17 items (MDF-fibro-17): evidence from validity, reliability and transcultural invariance between Portugal and Brazil

Dear Section Editor

My colleagues and I would like to thank you for the opportunity to resubmit our manuscript to the Journal of Clinical Medicine. The reviewer comments and our point-by-point responses are attached to this letter. We have also included an updated version of our manuscript with all the changes highlighted using the track-change option in Microsoft Word.

If you require any additional information, please do not hesitate to get in touch with us.

Reviewer

Thank you very much for your consideration of our comments and suggestions. I think the manuscript has improved a great deal by your revision.

R: We appreciate your comments and important feedback provided in the first round of revision.

Generally, I still think the manuscript should be thoroughly proof-read for English grammar by an English native speaker. The wording still seems off to me and I detected some typos throughout the manuscript.

R: The entire manuscript was sent to an English teacher for proof-read.

Content-wise, please make sure that the reader knows that FM is by definition predominantly defined as a disorder of widespread medically unexplained pain. Although fatigue is very important and also part of FM criteria, it is not the main defining symptom.

R: The reviewer is right. Thus, sentences were clarified.

Also, assessment of tender points has been omitted from the 2010 ACR criteria. If you used those criteria, you should consider not introducing and referring to tender points - or doing so less prominently.

R: We appreciate your concern and have revised some of the sentences referring to tender points.

Furthermore, I suggest using the term "persons with FM" or "patients with FM", not "FM carriers" or "patients of the disease".

R: The reviewer is right. We have revised the terms and followed your suggestion.